# The Glymphatic Response to the Development of Type 2 Diabetes

**DOI:** 10.3390/biomedicines12020401

**Published:** 2024-02-08

**Authors:** Edward D. Boyd, Li Zhang, Guangliang Ding, Lian Li, Mei Lu, Qingjiang Li, Rui Huang, Jasleen Kaur, Jiani Hu, Michael Chopp, Zhenggang Zhang, Quan Jiang

**Affiliations:** 1Department of Neurology, Henry Ford Health System, E&R B126, 2799 West Grand Boulevard, Detroit, MI 48202, USA; lzhang3@hfhs.org (L.Z.); gding1@hfhs.org (G.D.); lli2@hfhs.org (L.L.); qli1@hfhs.org (Q.L.); jkaur3@hfhs.org (J.K.); mchopp1@hfhs.org (M.C.); zzhang1@hfhs.org (Z.Z.); qjiang1@hfhs.org (Q.J.); 2Department of Radiology, Michigan State University, East Lansing, MI 48824, USA; 3Department of Public Health Sciences, Henry Ford Health System, Detroit, MI 48202, USA; mlu1@hfhs.org; 4Department of Physics, Oakland University, Rochester, MI 48309, USA; 5Department of Radiology, Wayne State University, Detroit, MI 48202, USA; jhu@med.wayne.edu; 6Department of Neurology, Wayne State University, Detroit, MI 28202, USA

**Keywords:** glymphatic system, magnetic resonance imaging, cerebrospinal fluid interstitial fluid exchange, immuno-histopathology, cognitive deficits

## Abstract

The glymphatic system has recently been shown to be important in neurological diseases, including diabetes. However, little is known about how the progressive onset of diabetes affects the glymphatic system. The aim of this study is to investigate the glymphatic system response to the progressive onset of diabetes in a rat model of type 2 diabetic mellitus. Male Wistar rats (*n* = 45) with and without diabetes were evaluated using MRI glymphatic tracer kinetics, functional tests, and brain tissue immunohistochemistry. Our data demonstrated that the contrast agent clearance impairment gradually progressed with the diabetic duration. The MRI data showed that an impairment in contrast clearance occurred prior to the cognitive deficits detected using functional tests and permitted the detection of an early DM stage compared to the immuno-histopathology and cognitive tests. Additionally, the quantitative MRI markers of brain waste clearance demonstrated region-dependent sensitivity in glymphatic impairment. The improved sensitivity of MRI markers in the olfactory bulb and the whole brain at an early DM stage may be attributed to the important role of the olfactory bulb in the parenchymal efflux pathway. MRI can provide sensitive quantitative markers of glymphatic impairment during the progression of DM and can be used as a valuable tool for the early diagnosis of DM with a potential for clinical application.

## 1. Introduction

Our traditional understanding of the role and hydrodynamics of the cerebrospinal fluid (CSF) has been fundamentally altered by the discovery of the glymphatic system [1,2,3,4,5]. The glymphatic system is a brain waste clearance system, providing a well-organized convective fluid flow that drives a substantial portion of the subarachnoid CSF to flow into the brain parenchyma through para-arterial pathways [1,2,3,4,5]. Then, the CSF mixes with the interstitial fluid (ISF), evoking CSF–ISF exchange, and both are cleared together along para-venous pathways with any associated solutes [3]. The dysfunction of glymphatic clearance is involved in the development and progression of neurodegenerative conditions, including Alzheimer’s disease, stroke, diabetes-associated brain damage, and traumatic brain injury (TBI) [3,5,6,7].

Type 2 diabetes mellitus (T2DM) is a common metabolic disease in middle and older age, affecting about one in five individuals over the age of 65 years. Numerous neurovascular and metabolic factors, such as cerebral vascular complications, decreased neurogenesis, oxidative stress and neurotoxicity, disturbances in glucose homeostasis, and defects in amyloid metabolism, have been implicated in the development of cognitive abnormalities [8]. However, the pathological mechanisms underpinning DM-induced cognitive impairments remain unclear. Moreover, although clinical trials demonstrate that medications that control blood glucose levels can significantly improve diabetes management, they do not prevent cognitive decline progression [9,10]. Thus, it is imperative to investigate the pathophysiological mechanisms underlying DM-induced cognitive decline in the aged population in an effort to develop therapies for this patient population.

We have studied the effects of diabetic mellitus (DM) on the glymphatic system [7]. Our studies have demonstrated that glymphatic measurements are sensitive to diabetic-induced changes in brain waste clearance [7]. A persisting question, however, is whether glymphatic measurements can provide information reflecting the progression of DM from the early stage to the more severe late stage. Thus, in this study, we investigate the sensitivity of MRI glymphatic measurements compared to immuno-histopathological analysis and functional behavioral tests at the early stages of DM and by the time of progression to severe diabetic conditions.

## 2. Materials and Methods

All the experimental procedures were conducted and performed in accordance with guidelines of the National Institutes of Health (NIH) for animal research under a protocol approved by the Institutional Animal Care and Use Committee of Henry Ford Hospital and the experimental guidelines of ARRIVE (items 8 and 10 to 13). All experimental procedures were approved by the Institutional Animal Care Committee of Henry Ford Hospital.

### 2.1. Animal Model and Groups

Male Wistar rats, 13 months of age (*n* = 54), were subjected to intraperitoneal (IP) injection of 210 mg/kg of nicotinamide (NTM) and 60 mg/kg of streptozotocin (STZ). The doses of NTM and STZ are slightly modified from the originally described method by Masiello et al., which has been demonstrated to produce non-insulin-dependent diabetic mellitus (DM) syndromes that resemble human type 2 diabetes [11,12]. This animal model mimics some of the pathological manifestations of human T2DM, including an increase in blood glucose, insulin secretory dysfunction, and glucose intolerance, but not hyperinsulinemia and obesity [11,13,14]. The diabetic state was assessed bi-weekly after STZ injection and monthly thereafter using measurements of the non-fasting plasma glucose concentration. These rats develop hyperglycemia (non-fasting plasma glucose concentrations > 200 mg/dL) starting at 2 weeks after STZ-NTM administration, which resembles the clinical manifestation of T2DM [15,16]. The DM rats also exhibited the progression of cognitive decline starting at 2 months (2M) after STZ-NTM injection.

A total of 54 rats were assigned into 1-month (1M, *n* = 18) and 3-month (3M, *n* = 18) DM and non-diabetes (non-DM, *n* = 18) groups. The 1M DM animals were at the early stage of diabetes, and we used this group of animals to test whether glymphatic measurements are sensitive to an early-stage diabetes condition. The MRI measurements were performed in 10 non-DM, 10 1M-DM, and 10 3M-DM rats. The 1M-DM, 3M-DM and non-DM rats were also subjected to functional behavioral tests (MWM and odor recognition tests, *n* = 8/group) and immuno-histological measurements (fibrin, MMP9, platelet, Aβ, and AQP4, *n* = 4/group).

### 2.2. MRI Measurements

The MRI measurements were obtained using a 7T scanner with a birdcage-type coil as the transmitter and a quadrature half-volume coil as the receiver. During the MRI measurements, anesthesia was maintained using medical air (1.0 L/min) with isoflurane (1.0–1.5%). Stereotactic ear bars were used to minimize movement, and the rectal temperature was maintained at 37 ± 1.0 °C using a feedback-controlled water bath.

In each MRI session, a fast gradient echo imaging sequence was used for the positioning of the animal in the magnet, and a spin echo sequence of T_2_-weighted imaging (T2WI) was acquired for the detection of brain tissue changes in the DM rat. The T2WI contained six sets of images using echo times of 15, 30, 45, 60, 75, and 90 ms; a repetition time of 4.5 s; a 32 × 32 mm^2^ field of view (FOV); a 128 × 128 matrix; and a 1 mm slice thickness of 13 slices.

The dynamic CSF–ISF exchange was measured using 3D T1-weighted MR images (T1WIs) with a Gd-DTPA contrast agent followed by a contrast-enhanced MRI (CE-MRI) protocol [7]. The 3D T1WIs were obtained with an echo time of 4 ms, a repetition time of 15 ms, a flip angle of 15°, a 32 × 32 × 16 mm^3^ field of view (FOV), and a 256 × 192 × 96 matrix interpolated to 256 × 256 × 128 voxels (0.125 × 0.125 × 0.125 mm) for analysis. Dynamic 3D T1WIs were acquired continuously for 6 h with 3 baseline scans, followed by intra-cisterna magna (ICM) Gd-DTPA contrast (21 mM, Gd-DTPA, MW 1 kD) delivery via the indwelling catheter while the MRI acquisitions continued [7]. A total of 80 µL of the paramagnetic contrast agent was delivered intrathecally at an infusion rate of 1.6 µL per minute (total infusion time = 50 min). 

### 2.3. Brain Section Preparation and Immunohistochemistry Evaluation

The rats were euthanized using Ketamine/Xylazine and then transcardially perfused with heparinized saline. Immunohistochemistry was performed on brain coronal sections (8 μm thick) from paraffin-embedded tissue corresponding to −3.0 to −4.0 mm into the bregma. The following primary antibodies were used: goat anti-fibrin/fibrinogen (Accurate Chemical & Scientific, Carle Place, NY, USA; 1:1000), rabbit anti-thrombocyte (Inter-Cell Technologies, Hopewell, NJ, USA; 1:900), mouse anti-endothelial barrier antigen (EBA, Sternberger Monoclonals, Lutherville, MD, USA; 1:1000), rabbit anti-aquaporin-4 (AQP4, Abcam, Cambridge, UK; 1:1500), rabbit anti-beta Amyloid 1–42 (Aβ, Abcam, Cambridge, UK; 1:100), and mouse anti-MMP-9 (Chemicon, Temecula, CA, USA; 1:1000). Double immunofluorescence staining of EBA with the fibrin/fibrinogen and thrombocytes was performed to identify the vascular fibrin and platelet accumulation, respectively. AQP4 plays an important role in mediating CSF–ISF exchange and clearance [3]. Double immunofluorescence staining of EBA with AQP4 was performed to identify the peri-vascular AQP4 expression. Immunostaining of MMP9 (a key protease associated with BBB leakage) was used to detect BBB disruption. Immunostaining of Aβ was used to detect Aβ deposition. For quantification, each coronal section was digitized throughout the hippocampus using a 40× objective and using an MCID system. The numbers of vessels with fibrin and thrombocyte immunoreactivity were counted and were presented as a percentage of the vessels. The peri-vascular AQP4 immunoreactivity area was measured and expressed as a percentage of the total peri-vascular area. The numbers of vessels with MMP9 immunoreactivity and peri-vascular Aβ deposition were counted and were presented as the density of vessels relative to the scan area (mm^2^). 

### 2.4. Functional Outcomes

All functional outcome tests were performed by observers blinded to the group allocation. For the assessment of cognitive impairments, the rats’ spatial learning function was measured using a MWM test performed daily for 5 days before sacrifice [17,18]. Briefly, the experimental apparatus consists of a circular water tank (140 cm in diameter) which is placed in a large test room with many external cues (e.g., pictures, lamps, etc.) that are visible to the rats. A transparent platform (15 cm in diameter) is submerged 1.5 cm below the surface of the water at a random location within the northeast quadrant of the maze. For each trial, the rat was placed randomly at one of four fixed starting points (north, south, east, and west) and allowed to swim for a maximum of 90 s. The latency in finding the hidden platform and the time spent within the correct quadrant were recorded. Data are presented as the percentage of time spent within the correct quadrant relative to the total amount of time spent finding the platform.

The rats’ olfactory learning and memory were measured using an odor recognition test before sacrifice [19]. Briefly, identical round wooden beads (Woodworks Ltd., Haltom City, TX, USA) were placed in the cages of two individually housed donor rats for 1 week without a bedding change to acquire animal-specific odors (designated as N1 and N2). During the initial familiarization period (2 days before testing), 4 unscented wooden beads (designated as F1–F4) were introduced into the home cage, where the testing rat would be familiarized with the presence of the beads for 24 h. On the following day (1 day before testing), the 4 now-familiar beads (F1–F4) were removed for 1 h. After this 1 h period, 3 familiar beads (F1–F3) and a scented wooden bead with a novel odor (N1) were randomly placed into the home cage for three 1 min trials, with 1 min inter-trial intervals. The timing of the 1 min trial is initiated when rat begins approaching any bead. This procedure produces habituation to N1 while minimizing olfactory adaptation. The exploration time for each of the 4 beads was recorded by experimenters blinded to which beads were familiar or novel (beads were number-coded). The odor recognition memory test was performed 24 h after the novel odor habituation phase. Four beads, including two familiarized home cage beads F1 and F2, a recently familiarized bead N1, and a bead with a novel odor N2 were placed in the home cage, following the same procedure outlined for the habituation phase. The exploration times for each bead were recorded. The focus of the test was to assess the non-spatial social odor-based novelty recognition of the N2 bead and the overnight memory for the N1 bead. A good odor recognition memory was indicated by increased time spent exploring N2 than N1 and the other beads in the trial. Data are presented as the percentage of time spent on N2 relative to the total amount of time spent on all beads.

### 2.5. Data Analysis

The data analysis was performed in a blind fashion. The general MRI processing procedure consisted of brain extraction, correct head motion registration, and voxel-by-voxel conversion into the percentage of the baseline signal using SPM8 (http://www.fil.ion.ucl.ac.uk/spm/ (accessed on 1 March 2019)). Generally, the T1 images in this work include all the head tissues. Therefore, the first necessary step is to remove the non-brain tissues in order to reduce the computations and improve the performance of the following registration steps. To this end, for each case, first, we manually draw a mask on the standard deviation (STD) map (computed from standard deviation of the time courses) on each coronal section, one by one, in order to encompass the entire brain. Second, scan-to-scan misregistration caused by head movement was corrected through the rigid-body alignment of each scan to the time-averaged (mean) image. Third, to ensure that the voxel intensity represented the percentage change relative to the average baseline images, all images were subtracted and divided by the baseline average image. The signal changes measured in the T1-weighted MRIs over time in preselected anatomical areas were used to obtain the average time evolution curve (TEC) of the regional tissue uptake of the paramagnetic contrast agents. The T1-weighted averaged baseline images, as well as the contrast-enhanced T1-weighted MRIs, were used to anatomically guide the placement of the ROIs. From the TEC, the percentage residual intensity ($\frac{signal\;intensity\;at\;end\;of\;experiment\;minus\;baseline\;intensity}{maximum\;signal\;intensity\;experiment\;minus\;baseline\;intensity}\times 100$) was calculated for comparison between the DM and non-DM rats. We also fit the TECs to a two-phase model based on exponential curves [20] to derive the clearance time constant. The first phase models the accumulation of the CA in each voxel during the injection period. This phase was not investigated in this manuscript. However, the second phase, which models the clearance of the CA from each voxel, is modeled using an exponential decay, *I*(*t*) = *I_max_* (*e*^−(*t*−*tm*)/τ2^) for *t* > *t_m_*, in which *I_max_* is the maximum intensity accrued at *t_m_*. τ2 is the clearance time constant and is estimated from the data. The clearance time constant (τ2) is used in the current study, and the larger this value is, the slower the CA clearance from the tissue.

### 2.6. Statistics

The results are presented as mean ± standard errors. To detect the effect of DM on the glymphatic changes in the MRI clearance time constant and residual intensity, one-way analysis of variance (ANOVA) was performed by testing the overall group effect first, followed by subgroup analyses if the overall effect was significant at a *p*-value < 0.05. The subgroup analysis was considered exploratory if the overall group effect was not significant at the 0.05 level. 

## 3. Results

### 3.1. Body Weight Changes between Groups

The body weights were 628 ± 21, 655 ± 16, and 691 ± 18 in the 3M-DM, 1M-DM, and non-DM groups, and significant differences were detected between the 3M-DM and non-DM rats (*p* = 0.03).

### 3.2. Structural Appearance of CSF–ISF Exchange with Progression of DM 

Dynamic MRI measurements can provide a visualization of glymphatic CSF–ISF influx and efflux changes affected by the progression of diabetes. Figure 1 shows the appearance of an MRI contrast over the five-hour infusion and clearance period in a 1M-DM, 3M-DM, and non-DM rat. The dynamic T1WIs in the middle-aged non-DM rat clearly revealed the time-dependent anatomical routes of para-vascular influx and anatomic enhancement, from the Gd-DTPA arrival in the cisterna magna (non-DM, 0), the appearance in the pituitary (Pit) recess (non-DM, 15 min) and in the olfactory bulb (non-DM, 30 min), and the contrast enhancement in the brain (non-DM, 90 min, 3 h, 5 h). Compared with the middle-aged non-DM rat, the time-point-matched early-stage 1M-DM rat exhibits slightly increased hyperintensity areas, especially in the olfactory bulb (1M-DM, 30, 90 min). The more severe diabetic 3M-DM rat exhibits significantly increased hyperintensity in all the brain areas during the late stage (3M-DM, 90 min, 3 h, 5 h) of the measurements. 

### 3.3. Quantitative Evaluation of Glymphatic Changes with the Progression of DM

To compare the glymphatic changes within the subregions in the brain, time evolution curves (TECs) were plotted after the injection of Gd-DTPA inti the whole brain (Figure 2A) and olfactory bulb (Figure 2B) in the 3M-DM (*n* = 10), 1M-DM (*n* = 10), and non-DM (*n* = 10) groups. Within the whole brain (Figure 2A), the 1M-DM rats exhibited a higher (215% of control) percentage residual intensity compared with the non-DM rats, but this did not reach statistical significance (*p* = 0.16). The 3M-DM rats exhibited a significantly increased residual intensity (*p* < 0.02, 360% of control) in the whole brain compared with the non-DM rats. The overall DM effect on the percentage residual intensity is significant (*p* < 0.01). We also derived a Gd-DTPA clearance time constant, τ2, by fitting the TECs to a two-phase model [7]. The overall difference in τ2 among groups is not significant (*p* = 0.05). The Gd-DTPA clearance time constant was longer in the 3M-DM (*p* = 0.07, 198 min) and 1M-DM (*p* = 0.15, 177 min) rats compared to the non-DM (123 min) rats.

The glymphatic responses and time evolution curves exhibited regional differences between the 3M-DM, 1M-DM, and non-DM animals. In the olfactory bulb, the overall differences in the percentage residual intensity and clearance time constant among groups are significant (*p* < 0.017 and *p* = 0.02). The olfactory bulb (OB, Figure 2B) in the DM rats exhibited a statistically significant increase in the Gd-DTPA percentage residual intensity (3M-DM vs. non-DM, 248%, *p* < 0.0005; 1M-DM vs. non-DM, 188%, *p* = 0.001) and clearance time constant (3M-DM, 207% of non-DM, *p* < 0.0001; 1M-DM, 134% of non-DM, *p* = 0.02) compared to in the non-DM animals.

### 3.4. Immunohistological Evaluation of the Progression of DM

The quantitative ex vivo histological findings obtained from the 3M-DM, 1M-DM, and non-DM groups are summarized in Table 1. Neurovascular dysfunction in the hippocampus has been implicated in DM-associated cognitive decline in the aged population [21,22,23]. In the current study, we found fibrin and platelet deposition within the cerebral blood vessels (thrombosis) mainly in the hippocampus, which was increased from the non-DM (fibrin, 4.99; platelet 3.66) to the 1M-DM (fibrin, 9.64, *p* = 0.08; platelet 7.31, *p* = 0.09) and then to the 3M-DM (fibrin, 11.79, *p* = 0.02; platelet 12.19, *p* < 0.01) groups after the NTM-STZ injection (Table 1, Figure 3A,B). In addition, the microvascular MMP9 expression increased gradually in the 1M group (6.04) and even more so in the 3M (9.16) group after NTM-STZ injection compared to the non-DM group (Table 1, Figure 3D). Moreover, a loss in the mean para-vascular aquaporin-4 (AQP4, Figure 3C) immunoreactivity was detected with the progression of DM (1M, 26.46; 3M, 16.50; non-DM, 29.92), but only showed a trend of statistical significance (*p* = 0.09) for the 3M-DM group compared with the non-DM group. Coincidentally, the peri-vascular beta-amyloid accumulation increased at 3 months (1M, 1.79, *p* = 0.45; 3M, 3.01, *p* = 0.15; non-DM 1.20) after NTM-STZ injection (Table 1, Figure 3D) but without a statistically significant difference. Collectively, our data suggest that DM-induced microthrombosis and microvascular disruption events became evident at a more advanced stage of DM (3 months after NTM-STZ injection), which was accompanied by a loss in peri-vascular AQP4 and beta-amyloid accumulation.

### 3.5. Cognitive Evaluation of the Progression of DM

To examine cognitive function with the progression of DM, spatial learning function and olfactory learning and memory were evaluated by means of MWM and odor recognition tests (Figure 4). The overall difference in the percentage of time is significant (*p* < 0.01) among different groups. Rats spent a smaller % of time (1M-DM, 51%, *p* = 0.14; 3M-DM, 44%, *p* < 0.01) in the correct quadrant of the MWM with the progression of DM compared to the non-DM rats (55%). In addition, compared to the non-DM rats (69%), rats also spent a much smaller percentage of time (1M-DM, 59%, *p* = 0.07; 3M-DM, 44%, *p* < 0.01) on a novel object with the progression of DM as measured using the odor recognition test. 

## 4. Discussion

Diabetes affects more than 240 million people worldwide, but the early detection of diabetes-associated neurodegeneration is still challenging [24,25,26,27]. In the current study, we demonstrated that there are alterations in the MRI markers of brain glymphatic measurements not only at the advanced stage of diabetes but also at an early stage of DM prior to apparent histological neurovascular abnormalities and cognitive decline. MRI of the glymphatic system is sensitive not only to early changes during the early stages of DM but also during the progression to an advanced stage of diabetes and thus could serve as an early diagnostic marker for DM-associated neurovascular damage and cognitive decline. 

In the current study, we used a nicotinamide- and streptozotocin-induced diabetic model in middle-aged rats which resembles the clinical manifestations of T2DM [11,12,16]. An advantage of this DM model is that the severity of the progression of DM is proportional to the time since the STZ-NTM injection [28]. By using this model, we are able to test the sensitivity of diagnostic markers at an early stage (1M) between immuno-histopathological analysis, functional behavioral tests, and MRI glymphatic measurements. MRI glymphatic measurement of the residual intensity exhibited a statistically significant increase in the OB for the early-stage DM group compared with the non-DM group. We found that the detection sensitivity for early-stage DM has a brain regional dependence and MRI glymphatic measurement in the OB region is more sensitive than that of the whole brain. The OB is a major solute efflux pathway of the glymphatic system [29,30,31]. The efflux of CSF with brain waste occurs along the OB across the ethmoid plate to reach the lymphatic network associated with the nasal mucosa [32,33]. Macromolecules [33,34] and immune cells [35,36] appear to exit the CNS through this route. A possible reason for the improved sensitivity to the residual intensity in the OB at an early DM stage may be attributed to the important role of the OB in the parenchymal efflux pathway. 

Our results support the initial understanding of glymphatic fluxes [1,2,3,4,5] and show that the MRI tracer is transferred through the para-arterial pathways by the bulk flow to reach major influx nodes in both DM and non-DM animals. The T2DM rats showed an increased residual intensity and clearance time constant with the progression to a severe stage of DM compared with age-matched non-DM rats. Both the residual intensity and clearance time constant provide statistically significant parameters to detect for early-stage DM in the OB compared with non-DM animals. Moreover, the advanced 3M-DM animals exhibited statistically significant and large differences in both parameters in the OB compared with the non-DM group. With the progression to a severe stage of DM compared to the non-DM rats, both the accumulation and clearance of the tracer became slower (Figure 2). Thus, MRI glymphatic measurements are sensitive to the progression of DM.

One of the strongest risk factors for the development of many neurodegenerative diseases, such as AD and diabetes, is advancing age. Diabetes also significantly contributes to the development of cognitive impairment and AD [37,38,39,40,41,42,43,44]. Moreover, recent findings have revealed multiple pathological similarities between the brains of AD and DM patients. Especially, similar aberrant insulin signaling between the brains of AD and DM patients supports the idea that AD can be considered “type III DM” [45,46,47,48], and the histopathological evidence confirms this [49,50,51,52,53]. The experimental induction of DM escalated the AD pathology, such as neurofibrillary tangle development and senile plaques, in several animal models [49,50,51,52,53], all of which has been linked to the development of cognitive abnormalities [8]. In the present study, we found that the pathological manifestations of neurovascular damage were detected at a more advance stage of DM mainly in the hippocampus, a region that is considered critical for cognition and highly vulnerable to DM- and aging-related pathology. Intriguingly, DM-induced vascular damage is associated with peri-vascular beta-amyloid accumulation, coinciding with the time when cognitive decline became apparent in the DM rats. Thus, our data suggest that vascular dysfunction and peri-vascular beta-amyloid accumulation may contribute to DM-induced cognitive impairments in aged rats. There is increasing awareness of age-related impairments in cognitive function and an imperative need to investigate the cerebral microvascular and associated glymphatic mechanisms underlying cognitive decline [7,54,55,56,57]. Our data showed that glymphatic measurements can detect early pathophysiological dysfunction at the early stage of DM compared with vascular evaluations of its pathology and functional tests. Glymphatic system impairment and the associated accumulation of molecular waste in the para-vascular space can potentially initiate a series of inflammatory responses that lead to neurovascular disorders, such as an enlarged para-vascular space [7], a pathological feature in the brain of DM and other dementia patients [58]. Therefore, the cumulative Aβ deposition and neurovascular disruption following glymphatic impairment may promote a positive feedback loop that more negatively affects the glymphatic system’s function in the DM-affected brain. However, the details of the relationship between the impairment of the glymphatic system, vascular damage, Aβ-mediated neuropathology, and cognitive deficits after DM warrant further investigation.

AQP4 plays an important role in the glymphatic system to mediate CSF–ISF exchange and clearance [3]. In the current study, the mean para-vascular aquaporin-4 (AQP4) immunoreactivity gradually decreased with DM progression and exhibited a statistically significant trend for the 3M-DM group compared with the non-DM group. Previous studies have demonstrated the dependency of parenchymal glymphatic clearance on AQP4 water channels in different neurological diseases [3,5,55,59,60]. The glymphatic transport of waste solutes, including Aβ, was significantly reduced in AQP4^−/−^ mice when compared to controls [3,5,55,60]. Our data showed the trend of a reduction in the AQP4 expression with an increased severity of DM, and a clinical investigation of idiopathic normal pressure hydrocephalus (iNPH) patients also found an attenuated expression of peri-vascular AQP4 in iNPH, potentially contributing to slow waste clearance, impaired glymphatic circulation, and subsequent neurodegeneration [59]. Assessment of the details of the relationship between AQP4 and the solute transport patterns in the brain requires further study.

The current dynamic MRI protocol for glymphatic measurements, which was also used in our previous studies, could be improved [7]. Firstly, the infusion time of the MRI contrast agents and the total monitoring time can be significantly shortened, as demonstrated by the Benveniste group [61,62]. Secondly, glymphatic transport can be further improved by using a better anesthetics strategy; the glymphatic transport can be enhanced by 32% in rats anesthetized with dexmedetomidine plus low-dose (~0.6%) isoflurane when compared with isoflurane (~2%) alone [62].

Our data demonstrate that MRI could provide sensitive quantitative markers of glymphatic impairment not only for the progression of DM-associated neurovascular dysfunction and cognitive decline but also for early diagnosis during the early stages of DM. Further investigations are required to mechanistically relate glymphatic dysfunction to vascular damage, DM pathology, AQP4 deficits, and cognitive impairment.

## Figures and Tables

**Figure 1 biomedicines-12-00401-f001:**
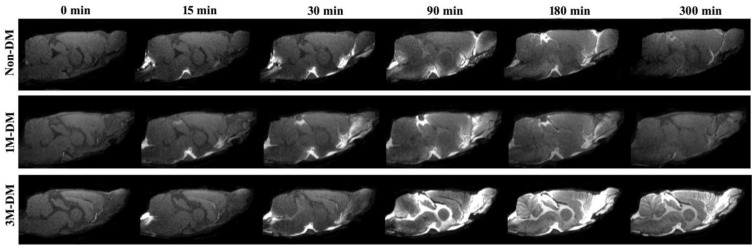
Dynamic tracer concentration changes between non-DM, 1M-DM, and 3M-DM rats: time series of MRI images demonstrates early influx (0–30 min) and anatomic enhancement 90 min, 3 h, and 5 h after ICM injection of Gd-DTPA in non-DM, 1M-DM, and 3M-DM animals.

**Figure 2 biomedicines-12-00401-f002:**
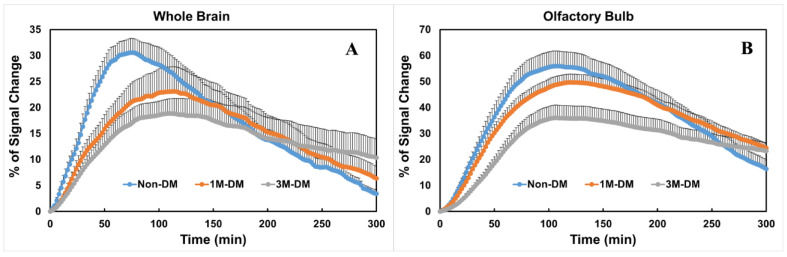
Quantitative time evolution differences in whole brain and olfactory bulb between non-DM, 1M-DM, and 3M-DM groups (*n* = 10 rats per group): MRI time evolution curves in the whole brain (**A**) and olfactory bulb (**B**) indicate that DM induced higher residual tracer concentrations and slower clearance rate than non-DM group. One directional error bar is used to show the standard error of the measurements within each group.

**Figure 3 biomedicines-12-00401-f003:**
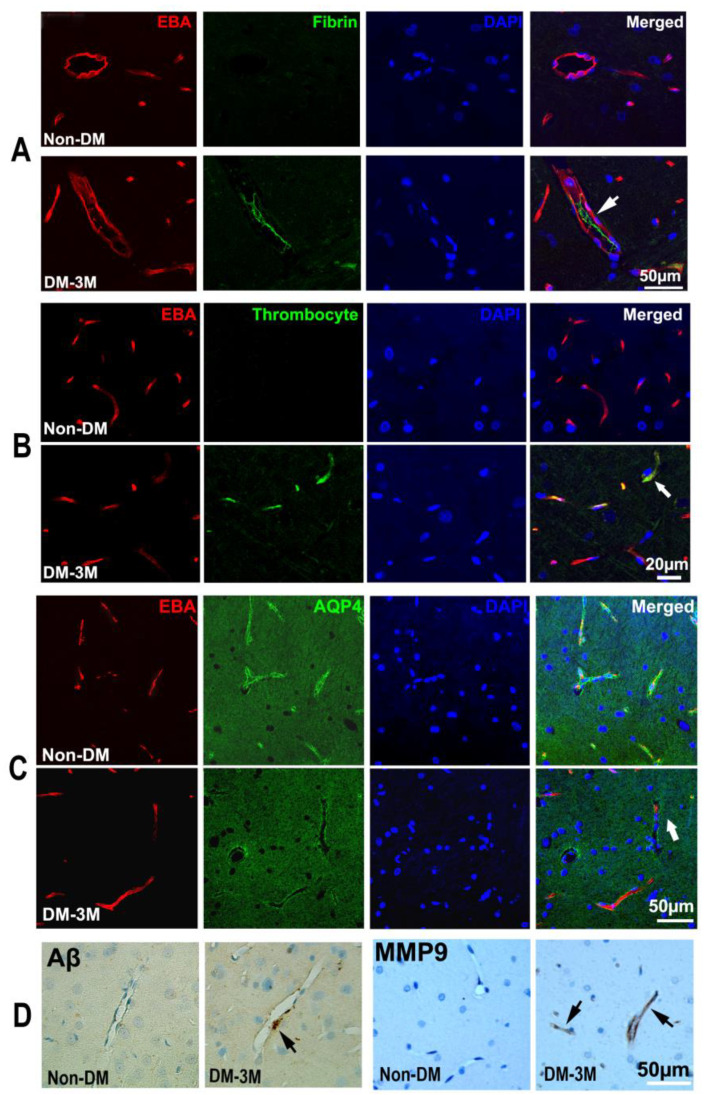
DM-induced vascular damage. The double immunofluorescent images show intravascular fibrin (green in (**A**), arrows) and platelet (green in (**B**), arrow) deposition and peri-vascular AQP4 immunoreactivity (green in (**C**), arrow) in the hippocampus of non-DM and 3M-DM rats. Peri-vascular beta-amyloid accumulation and MMP9 immunoreactivity were detected in 3M-DM ((**D**), arrows).

**Figure 4 biomedicines-12-00401-f004:**
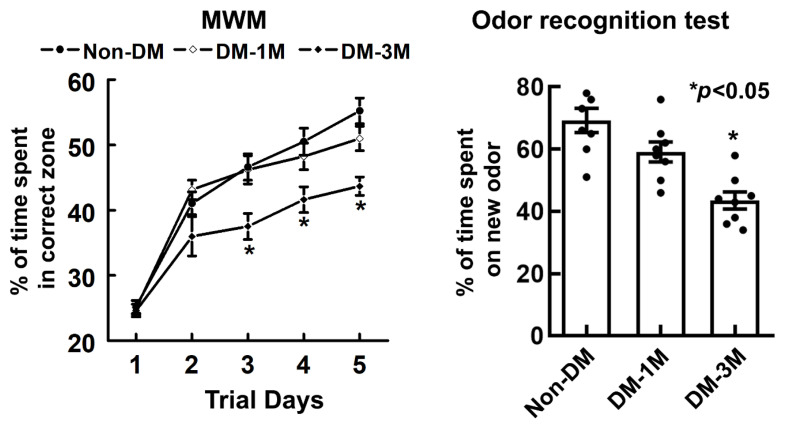
Results of evaluating cognitive function against the progression of DM. Spatial learning function and olfactory learning and memory were evaluated by means of MWM (**left**) and odor recognition (**right**) tests, respectively. (*n* = 18 rats per group). Values are mean ± SE.

**Table 1 biomedicines-12-00401-t001:** Histological measurements: mean ± SE (*n* = 18 rats per group).

Conditions	3M-DM	1M-DM	Non-DM
% AQP4 ^1^	16.50 ± 2.92	26.46 ± 2.31	29.92 ± 5.53
% MMP9 ^2,^*	9.16 ± 1.90 *	6.04 ± 0.92	3.15 ± 1.07
% platelet ^3,^**	12.29 ± 1.88 **	7.31 ± 1.49	3.66 ± 1.05
% Fibrin ^4,^*	11.79 ± 1.43 *	9.64 ± 1.43	4.99 ± 1.64
Aβ ^5^	3.01 ± 0.85	1.79 ± 0.27	1.20 ± 0.68

^1^ AQP4 is expressed in % of area. ^2^ MMP9 is expressed in % of blood vessels. ^3^ Platelet is expressed in % of blood vessels. ^4^ Fibrin is expressed in % of blood vessels. ^5^ Aβ is expressed in % of area. * = significant differences, *p* < 0.05; ** = significant differences, *p* < 0.01.

## Data Availability

The datasets used and/or analyzed during this study are available from the corresponding author upon reasonable request.

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
