# Peer review of "The Glymphatic Response to the Development of Type 2 Diabetes"

_biomedicines, 2024, doi:10.3390/biomedicines12020401_

Round 1

Reviewer 1 Report

Comments and Suggestions for Authors

The authors present an interesting study examining changes in the glymphatic system during the onset of diabetes. Briefly, the authors employ an established in vivo model of diabetes by conditioning Wistar rats with injection of nicotinamide and streptozotocin, before being tested after an allotted time frame had passed. Testing to see whether changes in glymphatic indices could be measured then proceeded in addition to changes in behavioural aspects. Based on the data presented, the authors imply that MRI could be used to detect onset of diabetes in the early stages based on changes in glymphatic activity.

In reviewing the manuscript, I made a number of observations. The following should be considered by the authors when preparing a suitable revision.

1.       Was the onset of diabetes confirmed in the rats who received the treatment? Body weight data is useful but were any other indices measured to confirm the efficacy of the treatment and the difference between animals?

2.       In figure 2, do the error bars represent replicate measurements in one animal, or do they represent the differences within a group of animals tested?

3.       For the behavioural tests of the animals, it would be useful to have the results graphed.

4.       It would be preferred if the n-numbers for each data set were stated in each figure/figure legend for clarity.

Reviewer 2 Report

Comments and Suggestions for Authors

Q1: 2 materials and methods

210 mg/kg of nicotinamide (NTM) and streptozotocin (STZ, 60 mg/kg),  

Your writing here is the same as that in your reference 7

I suggest it can be revised as

210 mg/kg of nicotinamide (NTM) and 60/kg streptozotocin (STZ)

Q2: According to your reference 11, their data showed that nicotinamide at 230 mg/kg, only blood glucose concentrations were significantly different (P < 0.05) from control values. Finally, the highest nicotinamide dosage (350 mg/kg) fully prevented STZ (65 mg/kg) -induced alterations.

Your reference 12 used nicotinamide 120mg/kg and STZ 60 mg/kg. Apparently, the doses of those medications were not the same as that in your work.

A modified protocol from reference11 and 12 may be better.

Q3: Did you test whether the use of STZ may directly affect brain?

Please see Front. Aging Neurosci., 18 May 2018 (https://doi.org/10.3389/fnagi.2018.00145)

https://www.frontiersin.org/articles/10.3389/fnins.2018.00653/full

Round 2

Reviewer 1 Report

Comments and Suggestions for Authors

The authors have suitably addressed my comments